# Oral Immunogenicity of Enterotoxigenic *Escherichia coli* Outer Membrane Vesicles Encapsulated into Zein Nanoparticles Coated with a Gantrez^®^ AN–Mannosamine Polymer Conjugate

**DOI:** 10.3390/pharmaceutics14010123

**Published:** 2022-01-04

**Authors:** Melibea Berzosa, Alzbeta Nemeskalova, Alba Calvo, Gemma Quincoces, María Collantes, Felix Pareja, Carlos Gamazo, Juan Manuel Irache

**Affiliations:** 1Department of Microbiology and Parasitology, Institute of Tropical Health, University of Navarra, 31008 Pamplona, Spain; mberzosa.1@alumni.unav.es (M.B.); nemeskal@faf.cuni.cz (A.N.); acalvoba@unav.es (A.C.); cgamazo@unav.es (C.G.); 2Department of Analytical Chemistry, University of Chemistry and Technology Prague, Technická 5, 166 28 Prague, Czech Republic; 3Department of Nuclear Medicine, Clínica Universidad de Navarra, 31008 Pamplona, Spain; gquinfer@unav.es (G.Q.); mcollant@unav.es (M.C.); fparejadelr@unav.es (F.P.); 4Department of Pharmacy and Pharmaceutical Technology, University of Navarra, 31008 Pamplona, Spain

**Keywords:** Enterotoxigenic *Escherichia coli* (ETEC), outer membrane vesicle (OMV), nanoparticles, oral vaccine, Gantrez, mannosamine

## Abstract

Enterotoxigenic *Escherichia coli* (ETEC) represents a major cause of morbidity and mortality in the human population. In particular, ETEC infections affect children under the age of five from low-middle income countries. However, there is no licensed vaccine against this pathogen. ETEC vaccine development is challenging since this pathotype expresses a wide variety of antigenically diverse virulence factors whose genes can be modified due to ETEC genetic plasticity. To overcome this challenge, we propose the use of outer membrane vesicles (OMVs) isolated from two ETEC clinical strains. In these OMVs, proteomic studies revealed the presence of important immunogens, such as heat-labile toxin, colonization factors, adhesins and mucinases. Furthermore, these vesicles proved to be immunogenic after subcutaneous administration in BALB/c mice. Since ETEC is an enteropathogen, it is necessary to induce both systemic and mucosal immunity. For this purpose, the vesicles, free or encapsulated in zein nanoparticles coated with a Gantrez^®^–mannosamine conjugate, were administered orally. Biodistribution studies showed that the encapsulation of OMVs delayed the transit through the gut. These results were confirmed by in vivo study, in which OMV encapsulation resulted in higher levels of specific antibodies IgG2a. Further studies are needed to evaluate the protection efficacy of this vaccine approach.

## 1. Introduction

Enterotoxigenic *Escherichia coli* (ETEC) belongs to intestinal *E. coli* pathotypes that cause acute diarrhea [1]. After ingestion, ETEC reaches the upper area of the human small intestine where it crosses the mucus layer and adheres to the epithelial cells through adhesins and colonization factors. Then, ETEC releases a heat-stable toxin (ST) or/and heat-labile toxin (LT), which both provoke a continuous release of water and electrolytes from the host cells, thus producing acute diarrhea [2].

Diarrhea disease caused by ETEC infection mainly affects children under the age of five from lower-middle income countries, resulting in about 300,000 to 500,000 deaths per year and 8,338 disability-adjusted life-years (DALYs) lost over one year [3]. Furthermore, it is the main cause of traveler’s diarrhea [4].

Nowadays, there is no licensed vaccine against ETEC. Thus, the development of an effective one is a WHO priority [5,6] Current vaccine approaches are based mainly on inducing protective immunity by the identification of key virulence factors of the pathogen. However, this strategy makes ETEC vaccine development challenging since this pathotype expresses a wide variety of antigenically diverse virulence factors that vary across regions and populations and over time [7].

Previous reports highlight the potential of vaccines based on outer membrane vesicles (OMVs). OMVs are released from the outer membrane of Gram-negative bacteria such as ETEC. These vesicles contain different bacterial components, including lipopolysaccharide (LPS), DNA, RNA, enzymes, peptidoglycan, virulence factors and pathogen-associated molecular patterns (PAMPs) [8]. Four main advantages make them suitable for vaccination: (i) adequate size (20–250 nm), which allows them to be recognized by the immune system, (ii) incorporation of a wide variety of antigens, thus reducing the probability of pathogen variants in all targets, (iii) safe profile and (iv) capability to both stimulate cellular and antibody-mediated immunity as well as mucosal immune responses [8,9]. There is currently one licensed OMV-based vaccine against Neisseria meningitides B (Bexsero^®^) [10,11].

Vaccines against enteropathogens should induce both a systemic and mucosal immune response in order to generate protective immunity [12]. For this reason, oral route of administration is a convenient strategy. Furthermore, oral vaccines are needle-free and can be administered by medical personnel without training, which can be a major social and economic advantage [13], especially considering that ETEC infections are prevalent mainly in lower-middle income countries. Despite these numerous advantages of oral vaccines, OMVs can be degraded and inactivated in the harsh gastrointestinal tract conditions before interaction with the mucosal immune system [14]. In order to minimize these drawbacks and improve the OMVs targeting to immune cells, one possible alternative may be their encapsulation in nanoparticles [13]. Moreover, the use of nanoparticles may also enhance OMV immunogenicity, minimizing the possibilities for inducing tolerance [15].

In the present work, OMVs obtained from two clinical ETEC isolates were encapsulated into zein nanoparticles, which were coated with a hydrophilic corona made of mannosamine–poly(anhydride) conjugate obtained by the covalent binding of mannosamine to the anhydride groups of Gantrez^®^ AN. These ETEC isolates were selected because they carried the desired enterotoxins, colonization factors (CFs) and non-classical virulence determinants genes, namely *eltAB, cfa/i*, *cs3*, *cs21*, *etpA*, *yghj, eatA* and *tibA*. Since these antigens are prevalent among ETEC strains and have been described as immunogenic [16,17,18,19,20], OMVs from these clinical isolates were considered optimal candidates to develop a broad-spectrum ETEC vaccine. In order to both protect OMV antigenicity from the deleterious gut conditions and facilitate their arrival to the intestinal epithelium, OMVs were nanoencapsulated in zein-based nanoparticles. Zein is the major storage protein of maize with a GRAS (generally regarded as safe) status. Moreover, zein can be easily transformed into biodegradable nanoparticles, without using toxic reagents [21,22]. Moreover, in this work, these nanoparticles were decorated with a polymer corona made from a Gantrez^®^–mannosamine conjugate to confer both mucus-permeating and immunostimulatory properties. On one hand, the hydrophilic character of the conjugate facilitates the diffusion of the nanoparticles through the mucus layer and their arrival to the intestinal epithelium [23]. In the latter, Gantrez^®^ AN possess an important capability to act as an active Th1 adjuvant through TLR exploitation [24], whereas mannosamine residues may facilitate the targeting of these nanocarriers for dendritic cells [25]. 

In the current work, OMVs containing important ETEC immunogens (LT, CFA/I, CS3, CS21, EtpA, TibA, YghJ and EatA) were selected to achieve a broad-protective vaccine. We demonstrated not only their antigenicity using ETEC infected patient’s sera but also their immunogenicity by intradermal route in the murine model. In order to administer them by the oral route, these OMVs were encapsulated into zein nanoparticles coated with the conjugate Gantrez–mannosamine in order to facilitate their arrival to the epithelium surface and the induction of a pro-Th1 immune response in both female and male mice.

## 2. Materials and Methods

### 2.1. Isolation of the OMVs

OMVs were obtained from two clinical isolates of ETEC (Manhiça Health Research Centre (CISM), Mozambique). Briefly, both strains were cultured in 500 mL of TSB under shaking overnight (37 °C, 125 rpm). Then, bacteria were inactivated by heat treatment in flowing steam (100 °C, 15 min). Cells were discarded by centrifugation at 6000× *g*, 20 min, 4 °C, and the supernatant filtered through a 0.22 μm filter (Corning^®^) and purified by tangential filtration using a 100 kDa concentration unit (Millipore). The retenate was frozen at −80 °C and subsequently lyophilized.

### 2.2. Characterization of OMVs

#### 2.2.1. Proteomic Analysis

Mass spectrometry (MS)-based proteome analysis was performed to identify OMV proteins from three independent batches. Samples were homogenized in lysis buffer (7 M urea, 2 M thiourea, 50 mM DTT) and protein digestion was performed as previously described [26]. MS analysis of the resulting peptides were performed in a Sciex 5600 Triple-TOF system (Sciex, Framingham, MA, USA) as described elsewhere [27]. The MS/MS data acquisition was performed using Analyst 1.7.1 (Sciex) and spectra files were processed through Protein Pilot Software (v 5.0.1-Sciex) using ParagonTM algorithm (v 5.0.1) for database search [28], ProgroupTM for data grouping and compared against the concatenated target-decoy UniProt proteome database (ETEC plus intestinal *E. coli* pathotypes). False discovery rate was evaluated using a non-lineal fitting method [29] and displayed results were those reporting a 1% global false discovery rate or lower. The peptide quantitation was performed using the Progenesis LC-MS software [26]. Proteins were quantified with at least two unique peptides.

#### 2.2.2. Immunoblotting

To determine the antigenic characteristics of HT, immunoblotting assay was performed [30]. In brief, the components separated electrophoretically were transferred from the gel to a nitrocellulose membrane (Whatman Protran^®^; Merck KGaA, Darmstadt, Germany, pore size 0.45 µm) using a semidry blotting system at 0.8 mA/cm^2^ for 30 min (Trans-Blot^®^ SD Transfer Cell, Bio-Rad, Hercules, CA, USA). Protein-binding sites were blocked with 5% skimmed milk in PBS at room temperature overnight. Next, the membranes were washed three times with PBS-Tween and incubated with eight different sera from healthy donors or eight different sera from patients infected with ETEC, all diluted 1:80. After incubation at room temperature for 4 h, the membranes were washed three times with PBS-Tween and treated with peroxidase (PO)-conjugated secondary antibody GAHu/IgG (H+L), HRP conjugate (1:1000) or GAHu/IgA (Fc), HRP conjugate (1:1000) for 60 min at room temperature. Finally, membranes were washed with PBS-Tween and the antibody–antigen complexes were visualized by addition of a substrate/chromogen solution (H_2_O_2_/4-chloro-1-naphthol).

### 2.3. Preparation of OMV-Loaded Nanoparticles

The encapsulation of OMVs into nanoparticles was performed by a two-step process as described previously [31,32]. In the first step, the conjugate of Gantrez^®^ AN and mannosamine (GM) was synthetized. For this purpose, 1 g Gantrez^®^ AN [poly(anhydride)] was dissolved in 120 mL of acetone. Then, 50 mg of mannosamine was added and the mixture was heated at 50 °C under magnetic agitation at 400 rpm for 3 h. Then, the mixture was filtered through a pleated filter paper and the organic solvent was eliminated under reduced pressure in a Büchi R-144 apparatus (BÜCHI Labortechnik AG, Switzerland) until the conjugate was completely dried. Finally, the resulting powder was stored at room temperature in a hermetically sealed container until use.

In the second step, the OMV-loaded zein nanoparticles (NPZ) were prepared following the displacement method [23] before coating by simple incubation with the GM conjugate. Briefly, 200 mg zein and 4 mg OMVs (2 mg of OMV from each ETEC strain) and 30 mg L-lysine were dissolved in 20 mL of 70% ethanol. Next, nanoparticles were obtained by the addition of 20 mL of water. The resulting nanoparticles were maintained under magnetic agitation for 10 min. Then, 1 mL of an aqueous solution of GM solution in water (10 mg/mL) was added to the suspension of OMV-loaded zein nanoparticles and incubated for 30 min. Finally, 400 mg of mannitol was added, and the mixture was dried in a Büchi R-144 spray-drier (BÜCHI Labortechnik AG, Flawil, Switzerland). The resulting nanoparticles were identified as OMV-GM-NPZ. Empty nanoparticles (GM-NPZ) were prepared in the same way as described above but in the absence of GM.

### 2.4. Characterization of Nanoparticles

The particle size, polydispersity index (PDI) and zeta potential were determined by photon correlation spectroscopy (PCS) and electrophoretic laser Doppler anemometry, respectively, using a Zetasizer analyzer system (Malvern^®^ Instruments, Malvern, UK). The diameter of the nanoparticles was determined after dispersion in ultrapure water (1 mg/mL) and measured at 25 °C by dynamic light scattering at 90° angle. The zeta potential was determined after dispersion in ultrapure water (2 mg/mL). The yield of the nanoparticles’ preparation process was determined as the difference between the initial amount of the polymer used to prepare nanoparticles and the weight of the freeze-dried carriers.

### 2.5. OMV Quantification

To determine the OMV loading and encapsulation efficiency, 30 mg OMV-GM-NPZ was dispersed in 1 mL deionized water and centrifuged at 15,000× *g*, 20 min. The supernatant (containing the non-encapsulated OMV fraction) was centrifuged using centrifuge tubes containing a 3 kDa filter (Amicon^®^) at 15,000× *g*, 20 min, to eliminate mannitol. Next, 40 μL supernatant was mixed with 40 μL sample buffer (Tris-HCl 62.5 mM, pH 6.8; 10% glycerol; 2% sodium dodecyl sulfate, SDS; 5% mercaptoethanol and bromophenol blue) and non-encapsulated OMV concentration was determined by SDS-PAGE and immunoblotting, using different concentrations of OMVs as reference pattern. Densitometry of each band was performed, and the values were obtained by computer-based densitometry using Image Studio Lite software (LI-COR, Lincoln, Nebraska, NE, USA). Briefly, protein-binding sites were blocked with 5% skimmed milk in PBS, overnight, room temperature. Next, membranes were washed three times with PBS-Tween and incubated with sera from rabbit hyperimmunized with OMV from ETEC H10407 (ATCC 35401), diluted 1:100. After incubation at room temperature for 4 h, the membranes were washed three times with PBS-Tween and treated with peroxidase (PO)-conjugated secondary antibody GAR/IgG (H+L), HRP conjugate (1:1000) for 60 min at room temperature. Finally, membranes were washed with PBS-Tween and the antibody–antigen complexes were visualized by addition of a substrate/chromogen solution (H_2_O_2_/4-chloro-1-naphthol). The calibration curves were performed by adding known quantities of OMVs, subjected to the same conditions as the samples to be analyzed. The OMV loading was expressed as the amount of OMVs (μg) per milligram of nanoparticles and the encapsulation efficiency (EE, expressed as a percentage) was calculated as the quotient between the amount of OMVs quantified and the total amount of vesicles added for the formulation of the nanoparticles.

### 2.6. Biodistribution Studies

Radiolabeling of OMVs and OMV-GM-NPZ was performed by technetium-99m (99m TcO4^−^) reduction with stannous chloride. Sodium pertechnetate was obtained by elution of a 99 Mo-99m Tc generator (10 GBq UTK, General Electric and Curium, London, UK) following the manufacturer’s instructions.

A total of 40 µL of SnCl_2_·2H_2_O (0.50 mg/mL to OMVs and 0.25 mg/mL to NP-OMVs) was used and no ^99m^Tc-tin colloids were produced during the radiolabeling reaction. OMVs (100 µg) and NP-OMVs (100 µg) were pre-tinned with a HCl acidified tin chloride solution, ^99m^TcO4^−^ in saline added and reduction carried out in a non-oxidizing atmosphere using He-purged vials and solutions. The radiochemical purity of radiolabeled OMVs and NP-OMVs was checked by thin layer chromatography (TLC) using ITLC-SG strips (Agilent Technologies, Folsom, CA, USA) developed with 0.9% NaCl. Radioactivity distribution was measured and quantified using a radioTLC system (iScan Bioscan, Washington, Columbia, USA). Radiolabeling proceeded with >95% yield, thus avoiding the need for further purification of the radiolabeled product.

After the quality control, 100 µg of the radiolabeled NP-OMVs was mixed with 8 mg of NP-OMVs.

Finally, radiolabeled OMVs (26.2 ± 1.1 MBq) and OMV-GM-NPZ (11.4 ± 0.5 MBq) were administered to nine-week-old female BALB/c mice (20 ± 1 g) (n = 4) by oral route to perform in vivo biodistribution images that were acquired at 1, 4, 7 and 10 h post-administration in a U-SPECT6/E-class (MILabs, Houten, The Netherlands) scanner using a UHR-RM-1mm multi pinhole collimator. Mice were placed prone on a multi-mouse scanner bed under continuous anesthesia with isoflurane (2 in 100% O_2_ gas) to acquire whole-body scans over 30 min. Following the SPECT acquisition, a CT scan was performed to obtain anatomical information. After the last image acquisition (10 h), animals were euthanized and gamma signal was detected ex vivo in a gamma-counter (Hidex, Turku, Finland) for the calculation of the percentage of injected dose (% ID/organ) in the following organs: stomach, small intestine, cecum and large intestine.

### 2.7. Mice Immunization and Specific Antibody Response

All mice were treated in accordance with institutional guidelines for treatment of animals (Protocol CEEA 027-20, University of Navarra). Nine-week-old female and male BALB/c mice (20 ± 1 g) were separated in six randomized groups of five animals and immunized with one single dose of OMVs, either free (100 μg in PBS) or encapsulated in nanoparticles (100 µg in PBS) by oral route, or with free OMVs (10 μg in PBS) by subcutaneous route.

Blood was taken before immunization (time 0) and at week 2, 3 and 4 post-immunization. Specific IgG2a antibodies against OMVs in sera were determined by ELISA assay. Briefly, 96-well plates (MaxiSorb; Nunc, Germany) were coated with 100 μL of 10 μg/mL OMVs in a coating buffer (60 mM carbonate buffer, pH 9.6). Unspecific binding sites were blocked with 3% bovine serum albumin (BSA) in PBS for 1 h at room temperature. Sera from the immunized mice were diluted 1:100 with 1% BSA in PBS and incubated for 4 h at room temperature. After five washes with PBS-Tween buffer, class-specific goat anti-mouse IgG2a (Sigma-Aldrich) conjugated antibody was added and incubated for 1 h at room temperature. The detection was carried out by incubating the sample with H_2_O_2_-ABTS™ substrate-chromogen for 15 min at room temperature. Absorbance was measured with an ELISA plate reader (Tecan, Männedorf, Switzerland) at a wavelength of 405 nm.

### 2.8. Statistical Analysis

All statistical significance analyses were carried out using parametric one-way ANOVA. *p* values of <0.05 were considered statistically significant. All calculations were performed using Prism7^®^ software (San Diego, CA, USA).

## 3. Results

### 3.1. Outer Membrane Vesicles Proteomic Analysis

Protein analysis showed the presence of 565 proteins in OMVs with at least 6 unique peptides. Among them, 293 proteins were identified with a cellular localization described in the UniProt Database. The most predominant localization was the membrane (30%), of which 11% of the proteins were located in the outer membrane (Figure 1). 

Proteome analysis demonstrated the presence of important ETEC immunogenic virulence factors, such as colonization factors, LT and non-classical factors (Table 1 and Table 2). Furthermore, we identified other common *E. coli* conserved antigens (e.g., FliC, Ag43, OmpA, OmpC or OmpX) and a total of 24% of proteins associated with other *E. coli* pathotypes (Enterohemorrhagic (EHEC), Enteroaggregative (EAEC), Enteropathogenic (EPEC) and Shiga toxin-producing (STEC)). For instance, we detected the virulence factor, AIDA, which is frequently found in Enterohemorrhagic *E. coli* (O157:O7).

### 3.2. Outer Membrane Vesicles Antigenicity

To investigate OMV antigenicity, their reactivity with sera from patients infected with ETEC and from healthy donors was determined by immunoblotting. Results indicated that proteins that are conserved among the Enterobacteriaceae family, such as OmpA, OmpW and OmpX, were recognized by IgG antibodies present in both healthy donors and in infected patient’s sera. However, the ETEC specific protein TibA showed reactivity only with infected patient’s sera, especially with IgA isotype (Figure 2 and Figure 3).

### 3.3. Characterization of OMVs-Containing Nanoparticles

Nanoparticles containing OMVs (OMVs-GM-NPZ) showed a mean size of 211 nm and a negative zeta potential of about −48.7 mV. The encapsulation efficiency was close to 70%, which corresponds to an OMV loading (payload) of 5.8 μg per mg nanoparticle. Table 3 summarizes the main physicochemical properties of the nanoparticles employed in this study.

### 3.4. Biodistribution of OMV-GM-NPZ Compared to Free OMV

To compare the biodistribution of free and encapsulated OMVs, the samples were labeled with technetium-99m and administered to BALB/c female mice (n = 4) by oral route. Image analysis performed 1, 4, 7 and 10 h post-administration indicated that encapsulation of the vesicles delayed the transit through the gastrointestinal tract (Figure 4). This was confirmed by technetium-99m quantification 10 h post-administration in stomach, small intestine, cecum and large intestine. Results showed higher technetium-99m levels in stomach, cecum and large intestine of mice treated with OMV-GM-NPZ compared to those treated with free OMVs (Figure 5).

### 3.5. Evaluation of the Immunogenicity of OMV-GM-NPZ

To investigate OMV-GM-NPZ immunogenicity, female and male BALB/c mice (n = 5) were immunized with free OMV (100 μg) by oral route, OMV-GM-NPZ (100 µg) by oral route or free OMV (10 μg) through subcutaneous administration. Specific serum levels of BALB/c mice IgG2a from mice immunized through subcutaneous route supported vaccine candidate immunogenicity (Figure 6). As expected, free OMV did not elicit specific antibody expression by the oral route, but their immunogenicity improved after their encapsulation. Figure 6 shows one female and two male mice as responders.

## 4. Discussion

Currently, there are no licensed vaccines against ETEC [5,33]. Several ETEC vaccine candidates have been proposed, some of them being under clinical investigation. The current approaches are based on CFs or/and enterotoxins attempting to interrupt intestinal colonization [7]. For instance, inactivated vaccines such as ETVAX are being considered. This candidate comprises a mixture of bacterins of ETEC strains expressing CFA/I, CS3, CS5, CS6 factors, the LCTBA LT toxoid and the double mutant dmLT [34]. Other examples include subunit vaccines, such as the CFA/I/II/IV- 3X STaN12S-dmLT multi-epitope fusion (MecVax) strategy [35]. Despite the fact that these approaches have been demonstrated to be immunogenic and able to elicit protection against ETEC infection in travelers (ETVAX) [36] or in a pig challenge model (MecVax) [35], a broadly protective vaccine has not yet been achieved.

The main goal of ETEC vaccine development is to target populations of low-middle income countries, especially children under five years of age [37]. In order to comply with this purpose, there are still challenges to overcome. These include the genetic plasticity of *E. coli* genomes in addition to the wide variety of virulence factors and the promiscuous adaptation, e.g., ~25 antigenically distinct antigens representing the colonization factors. This indicates the need for a multivalent vaccine strategy that is able to target the most highly conserved molecules [7]. In this way, we propose an OMV-based vaccine for its ability to carry numerous PAMPs and pathovar-specific antigens, whose immunogenicity have been demonstrated [38,39].

In the present work, bacterial vesicles were obtained from two ETEC clinical isolates from an endemic area of Mozambique. Their proteome analysis showed the presence of immunogenic conserved ETEC virulence factors, i.e., CFA/I, CS3, CS21, LT, EtpA, EatA, TibA and YghJ, all highly prevalent in low-middle income countries [40,41,42,43,44,45]. Furthermore, outer membrane proteins (OmpA, OmpC, OmpX, OmpW and OmpF), the flagellin FliC, autotransporter Ag43, fimH and fimG adhesins from fimbriae type I, the chaperone Skp and CexE protein were detected. All these proteins have also been proposed to further evaluate as possible vaccine candidates [37,46,47].

In addition to antigen detection in OMVs, antigenicity studies using sera from ETEC infected patients indicated the presence of IgG and IgA antibodies against TibA, OmpA, OmpW or OmpX. These results are consistent with the immune response elicited by the human experiment challenge with ETEC, carried out by Chakraborty et al. [48]. These observations support the use of OMV combination as a vaccine approach. Furthermore, in this study, we demonstrated the OMV immunogenicity in immunized BALB/c mice after subcutaneous administration, showing significant specific antibodies IgG2a levels compared to pre-immunization time.

The ETEC vaccines that are currently under clinical investigation are administered by parenteral or dermal routes, generally leading to systemic immune responses. However, enteropathogens such as ETEC colonize and invade the host at mucosal sites, and it is therefore necessary to induce mucosal immune responses to confer sufficient protection against these pathogens [13,49]. In this way, the selection of a suitable route of administration is of the utmost importance. In this work, the oral route was selected since it contributes to elicit intestinal immune response in addition to allowing for needle-free self-administration, thus constituting economic and social advantages [50,51,52]. However, oral immunization shows some challenges: (i) low vaccine absorbance due to its degradation by the acidic stomach pH and intestinal proteolytic enzymes [15]; (ii) time limitation, since the absorption of oral vaccines is limited to their residence time in the small intestine (3–4 h), which is the region where most of the absorption processes take place [53]; (iii) tolerogenic tendency of the mucosal tissues [54,55]. The encapsulation of the antigen complex with a suitable adjuvant should overcome these challenges [56,57]. In this study, OMVs were encapsulated into zein nanoparticles coated with a Gantrez–mannosamine conjugate (OMV-GM-NPZ). This hydrophilic corona allowed us to obtain nanoparticles with adjuvant properties, inducing Th1 immune responses through the receptors TLR2 and TLR4 [24,25,58,59,60]. Biodistribution studies indicated that the encapsulation of OMV into these nanoparticles delayed their transit through the gastrointestinal tract, suggesting that a prolonged residence would increase the possibilities for OMVs to reach the GALT. These data were confirmed by in vivo studies, which indicated that OMV encapsulation results in higher specific antibodies IgG2a levels.

In summary, we propose ETEC OMVs carrying important immunogens, such as pathovar-specific virulence factors (CFA/I, CS3, CS21, LT, EtpA, EatA, TibA and YghJ) and other conserved antigens (FliC, OmpA, Skp or Ag43), encapsulated into zein nanoparticles coated with a Gantrez–mannosamine conjugate as vaccine candidate against ETEC. The results showed in this work demonstrated the capability of the antigen complex to induce an immune response after subcutaneous immunization in the murine model. Furthermore, their encapsulation allows oral administration, eliciting specific antibodies IgG2a in immunized BALB/c mice with OMV-GM-NPZ. These results would be related to the capability of the developed nanoparticles to increase their residence within the gut, as evidenced in the biodistribution studies. Further studies are needed to evaluate the protection efficacy of this vaccine approach in an ETEC challenge.

## 5. Conclusions

5.1. ETEC OMVs carrying important immunogens demonstrated antigenicity using ETEC infected patient’s sera.

5.2. ETEC OMVs carrying important immunogens induce an immune response after subcutaneous immunization in BALB/c mice.

5.3. Their encapsulation into zein nanoparticles coated with a Gantrez–mannosamine conjugate (OMV-GM-NPZ) allow them to increase their residence within the gut when they are orally administered in the murine model.

5.4. BALB/c mice immunized with OMV-GM-NPZ by oral route showed specific antibodies IgG2a.

## Figures and Tables

**Figure 1 pharmaceutics-14-00123-f001:**
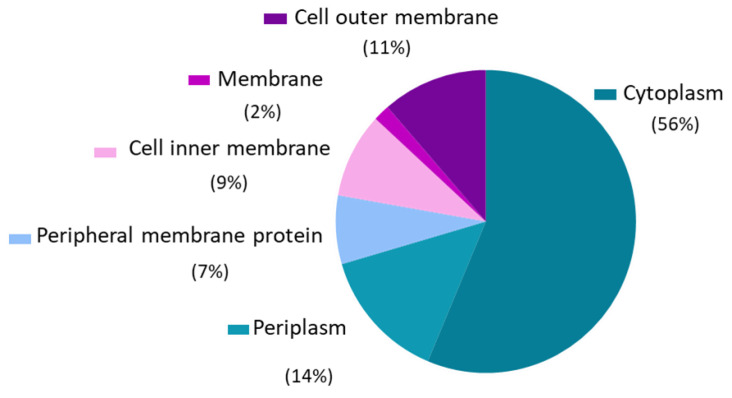
Subcellular distribution of proteins identified by proteomics of outer membrane vesicles (OMVs) isolated from two clinical Enterotoxigenic *Escherichia coli* (ETEC) isolates.

**Figure 2 pharmaceutics-14-00123-f002:**
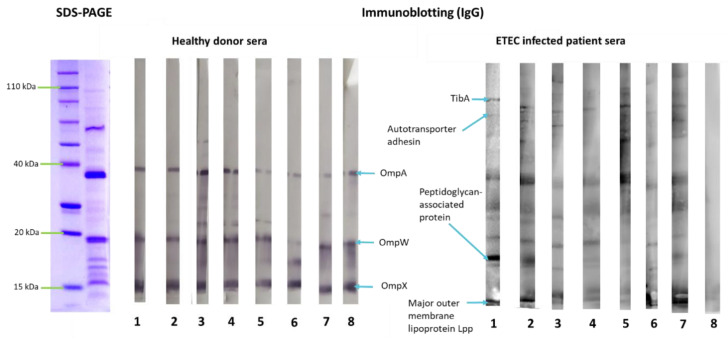
IgG reactivity with outer membrane vesicles (OMVs) isolated from two clinical Enterotoxigenic *Escherichia coli* (ETEC) isolates incubated with ETEC infected patient’s sera (n = 8) or healthy donor sera (n = 8). Protein-binding sites were blocked with 5% skimmed milk in PBS at room temperature overnight. After incubation with sera, the membranes were treated with peroxidase (PO)-conjugated secondary antibody GAHu/IgG (H+L), HRP conjugate (1:1000) for 60 min at room temperature. The antibody–antigen complexes were visualized by addition of a substrate/chromogen solution (H_2_O_2_/4-chloro-1-naphthol). Blue arrows indicate previously identified OMV proteins that were recognized by IgG antibodies.

**Figure 3 pharmaceutics-14-00123-f003:**
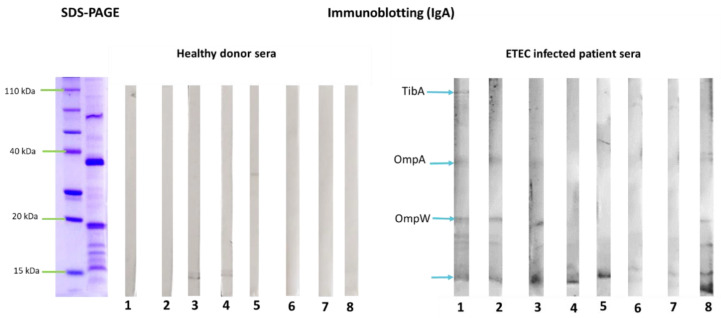
IgA reactivity with outer membrane vesicles (OMVs) isolated from two clinical Enterotoxigenic *Escherichia coli* (ETEC) isolates incubated with ETEC infected patient’s sera (n = 8) or healthy donor’s sera (n = 8). Protein-binding sites were blocked with 5% skimmed milk in PBS at room temperature overnight. After incubation with sera, the membranes were treated with peroxidase (PO)-conjugated secondary antibody GAHu/IgA (Fc), HRP conjugate (1:1000) for 60 min at room temperature. The antibody–antigen complexes were visualized by addition of a substrate/chromogen solution (H_2_O_2_/4-chloro-1-naphthol). Blue arrows indicate previously identified OMV proteins that were recognized by IgA antibodies.

**Figure 4 pharmaceutics-14-00123-f004:**
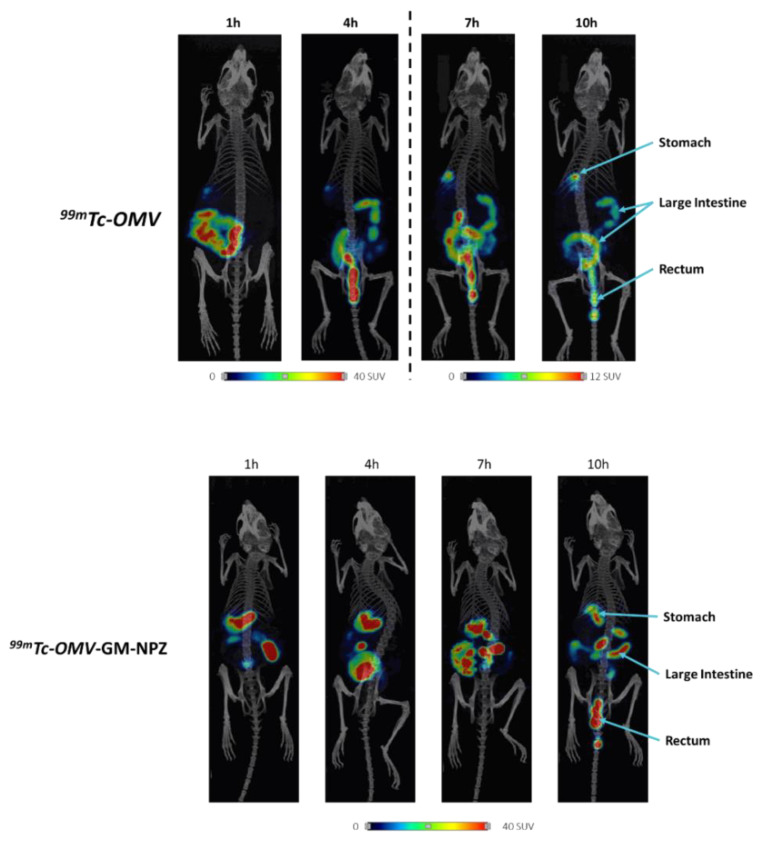
Biodistribution of radiolabeled outer membrane vesicles (OMV) or OMV encapsulated into zein nanoparticles coated with Gantrez^®^ (GM-NPZ) after oral administration in BALB/c mice. Location of the radiolabeled samples was obtained after 1, 4, 7 or 10 h using a microSPECT/CT model U-SPECT6/E-class (MILabs) apparatus. Samples were labeled with 99mTc and the intensity of the signal was expressed in Standardized Uptake Value (SUV). Blue arrows indicate organs where the signal was detected.

**Figure 5 pharmaceutics-14-00123-f005:**
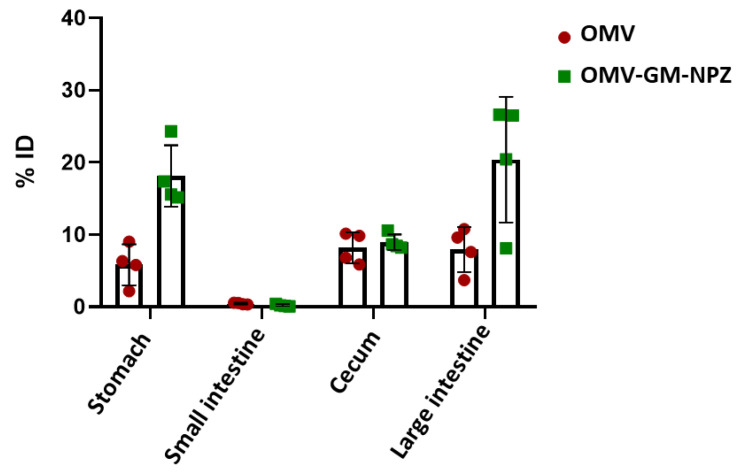
Biodistribution of radiolabeled OMV or OMV-GM-NPZ after oral administration in BALB/c mice. Percentage (%) of 99mTc activity in the different organs 10 h post-administration of OMV (100 μg) or OMV-GM-NPZ (100 μg).

**Figure 6 pharmaceutics-14-00123-f006:**
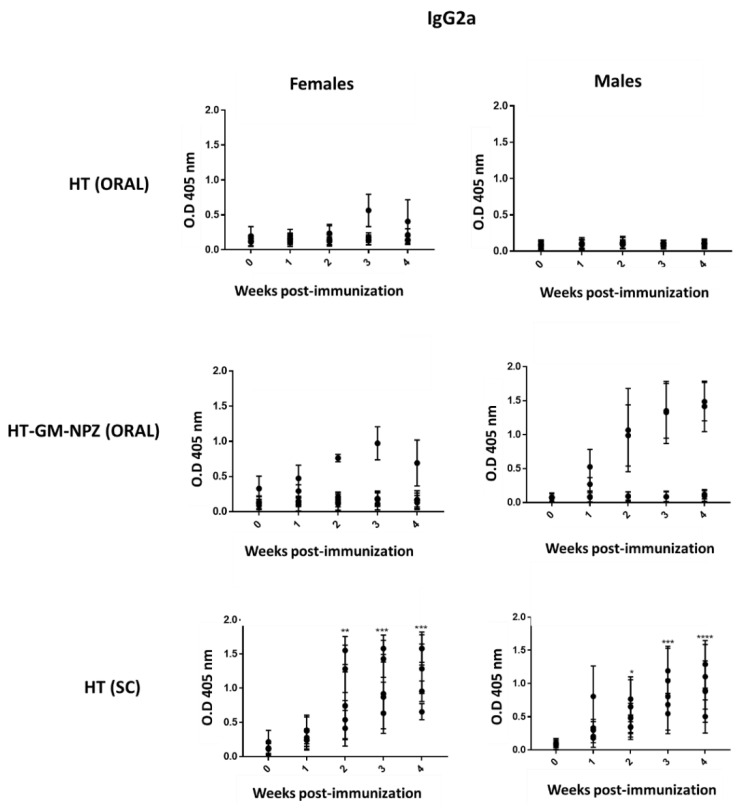
Antibody immune response induced after oral or subcutaneous (SC) vaccination of BALB/c mice with free outer membrane vesicles (OMV) or OMV encapsulated into zein nanoparticles coated with Gantrez^®^ (GM-NPZ). Specific serum IgG2a against OMV in immunized BALB/c mice with 100 μg of OMV or 16 mg of OMV-GM-NPZ taken from week 0 to 4 week after immunization. (* *p* < 0.05; ** *p* < 0.01; *** *p* < 0.001; **** *p* < 0.0001; vs. pre-immunization time). Error bars represent SD.

**Table 1 pharmaceutics-14-00123-t001:** Enterotoxigenic *Escherichia coli* (ETEC) virulence factors identified in outer membrane vesicles (OMV) isolated from the ETEC clinical isolate n°1. Protein quantification is expressed in iBAQ.

ETEC (Clinical Isolate n°1)			
UniProt Identifier	Gene (Strain)	Protein	iBAQ
D7GKK6	*lngA* (ETEC1392/75_p557_00053)	Putative pilus biosynthesis protein	6.50 × 10^8^
D7GKE0	*etpA* (ETEC1392/75_p1018_132)	Putative heamagglutinin afimbrial adhesin	1.91 × 10^8^
D7GK42	*eltB* (ETEC1392/75_p1018_007)	Heat-labile enterotoxin B chain	9.6 × 10^7^
E3PJ90	*yghJ* (ETEC_3241)	Putative lipoprotein YghJ (Putative lipoprotein AcfD homolog)	3.39 × 10^7^
D7GKA6	*cstA-H* (ETEC1392/75_p1018_087)	Putative CS3 fimbrial subunit	3.20 × 10^7^
Q9XD84	*tibA* (ETEC_2141)	Adhesin/invasin TibA autotransporter	6.72 × 10^6^
C8UFQ7	*yghJ* (ECO111_3795)	Putative lipoprotein AcfD homolog	5.49 × 10^6^
D7GK41	*eltA* (ETEC1392/75_p1018_006)	Heat-labile enterotoxin A chain	4.12 × 10^6^
E3PPC4	*cfaB* (ETEC_p948_0400)	CFA/I fimbrial subunit B (CFA/I antigen)	3.53 × 10^6^
B7UI41	*yghJ* (E2348C_3253)	Predicted inner membrane lipoprotein	3.50 × 10^5^
Q84GK0	*eatA* (ETEC_p948_0020)	Serine protease EatA	1.55 × 10^5^

**Table 2 pharmaceutics-14-00123-t002:** Enterotoxigenic *Escherichia coli* (ETEC) virulence factors identified in outer membrane vesicles (OMV) isolated from the ETEC clinical isolate n°2. Protein quantification is expressed in iBAQ.

ETEC (Clinical Isolate n°2)			
UniProt Identifier	Gene (Strain)	Protein	iBAQ
Q9XD84	*tibA* (ETEC_2141)	Adhesin/invasin TibA autotransporter	1.84 × 10^8^
D7GK42	*eltB* (ETEC1392/75_p1018_007)	Heat-labile enterotoxin B chain	5.97 × 10^7^
E3PPC4	*cfaB* (ETEC_p948_0400)	CFA/I fimbrial subunit B (CFA/I antigen)	3.84 × 10^7^
E3PJ90	*yghJ* (ETEC_3241)	Putative lipoprotein YghJ (Putative lipoprotein AcfD homolog)	2.68 × 10^7^
E3PPC6	*cfaE* (ETEC_p948_0420)	Cfa/I fimbrial subunit E	1.89 × 10^7^
D7GKK6	*lngA* (ETEC1392/75_p557_00053)	Putative pilus biosynthesis protein	1.70 × 10^7^
D7GK41	*eltA* (ETEC1392/75_p1018_006)	Heat-labile enterotoxin A chain	1.49 × 10^7^
C8UFQ7	*yghJ* (ECO111_3795)	Putative lipoprotein AcfD homolog	1.06 × 10^7^
E3PPC3	*cfaA* (ETEC_p948_0390)	CfA/I fimbrial subunit A	5.31 × 10^6^
Q84GK0	*eatA* (ETEC_p948_0020)	Serine protease EatA	2.55 × 10^6^
D7GKE0	*etpA* (ETEC1392/75_p1018_132)	EtpA adhesin	2.52 × 10^6^

**Table 3 pharmaceutics-14-00123-t003:** Physicochemical characterization of Enterotoxigenic *Escherichia coli* (ETEC) outer membrane vesicles (OMV)-containing nanoparticles. EE: encapsulation efficiency. Data expressed as mean ± SD (n > 3).

Formulation	Size (nm)	Polydispersity Index (PDI)	Zeta Potential (mV)	OMV Loading (μg/mg)	EE (%)
GM-NPZ	245 ± 6.00	0.07 ± 0.01	−51 ± 1.60	N.D	N.D
HT-GM-NPZ	211 ± 5.00	0.14 ± 0.04	−48.71 ± 0.68	5.80	~70%

N.D. No data.

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
