# Peer review of "Oral Immunogenicity of Enterotoxigenic Escherichia coli Outer Membrane Vesicles Encapsulated into Zein Nanoparticles Coated with a Gantrez® AN–Mannosamine Polymer Conjugate"

_pharmaceutics, 2022, doi:10.3390/pharmaceutics14010123_

Round 1
Reviewer 1 Report
Dear Editor,
In the current study the authors have investigated the use of outer membrane vesicles (OMVs) isolated from two ETEC clinical strains as vaccine candidate. Zein particles were used for the delivery and encapsulation. The scientific value of this study is high. The presentation of article is up to the standards of journal.
I have few minor comments.
In abstract please avoide using abbreviations.
In introduction, please give a brief intro of zein to make it more reader friendly for researchers from other branches.
Why Zein was selected for this study? please explain it in introduction.
What other alternatives are out there?, please give few examples, in introduction section.
Please write the aim and novelty of the study clearly. Include it at the end of the introduction.
Please cite the methodology of immunoblotting. Give a proper reference.
For the quantification of loading efficiency, which mathematical equations were used. Please indicate them in methodology.
Did the authors checked the release behavior of vaccine from particles? if not, please check it in a simulated environment and give the release pattern. For reference, please look at the below article:
https://www.sciencedirect.com/science/article/pii/S0928493116313522.
Please provide a brief conclusion of the study.
Author Response
Thank you very much for reviewing our manuscript. In the attached word document you will find our point-by-point response to the reviewer´comments.

Reviewer 2 Report
In this study, free, OMVs or OMVs encapsulated into zein nanoparticles covered with a Gantrez -mannosamine conjugate were administered by oral route. These results were confirmed by in vivo study, which indicated that OMV encapsulation results in higher specific antibodies IgG2a levels. However, there are still many areas that need to be improved. Therefore, major revision is needed to further improve this manuscript. After several additional modification, this work can be published.
- The authors should provide SEM and TEM images of OMV-Loaded nanoparticles.
- Are OMV-Loaded nanoparticles stable in the stomach and intestines?
- It is recommended that the author provide a schematic diagram of this work.
- Are OMV-Loaded nanoparticles stable in the body? Will it degrade?
- Some relevant references are suggested to be cited (Chemical Society Reviews 49.11 (2020): 3244-3261; Chemical Society Reviews, 2021, 50(15): 8669-8742).
Author Response

(The authors gave the same response as above.)
